# Metabolic dysfunction-associated steatotic liver disease (MASLD) can be a possible predictive factor of incident CKD: NAGALA cohort study

Yuriko Ono[1], Kimiko Sakai[1], Takuro Okamura[1], Hiroshi Okada[1], Akihiro Obora[2], Takao Kojima[2], Masahide Hamaguchi[1]*, Michiaki Fukui[1]

**1** Department of Endocrinology and Metabolism, Kyoto Prefectural University of Medicine, Graduate School of Medical Science, Kyoto, Japan, **2** Department of Gastroenterology, Asahi University Hospital, Gifu, Japan

* mhama@koto.kpu-m.ac.jp

## Abstract

### Background

A decade-long follow-up study identified metabolic dysfunction-associated fatty liver disease (MAFLD) as an independent predictor for the onset of chronic kidney disease (CKD). In 2023, a Delphi consensus introduced the term metabolic dysfunction-associated steatotic liver disease (MASLD) as an updated nomenclature. This study aims to evaluate whether MASLD, as the newly defined concept of steatotic liver disease, functions as an independent risk factor for CKD development. Additionally, the study seeks to examine the association between MASLD and CKD, while identifying contributing risk factors.

### Methods

This research involved a retrospective cohort study conducted on individuals participating in health checkups at Asahi University Hospital, Japan, from 1994 to 2023. Logistic regression analysis was employed to investigate the relationship between MASLD and the incidence of CKD over a five-year follow-up period.

### Results

A total of 15,873 participants were included in this study. The incidence of CKD was highest among individuals with MASLD (9.5%). Multivariate analysis demonstrated that MASLD was significantly associated with an increased risk of CKD, with an odds ratio (OR) of 1.37 (95% CI 1.12–1.67, p = 0.002). Additional factors such as age (OR 1.04, 95% CI 1.03–1.05, p < 0.001) and estimated glomerular filtration rate (eGFR) (OR 0.88, 95% CI 0.87–0.89, p < 0.001) were also identified as significant predictors of CKD. These findings suggest a robust association between MASLD and

**Data availability statement:** Due to legal and ethical restrictions imposed by the Asahi University Hospital Ethics Review Committee and by Japan's Ethical Guidelines for Medical and Biological Research Involving Human Subjects and the Act on the Protection of Personal Information (APPI), the individual-level dataset underlying this study cannot be shared publicly. Qualified researchers may request access to a de-identified dataset sufficient to reproduce the results by contacting the Data Access Committee (DAC), Asahi University Hospital (postal address: Data Access Committee [c/o Department of Gastroenterology], Asahi University Hospital, 3-23 Hashimoto-cho, Gifu-shi, Gifu 500-8523, Japan; Tel. +81-58-253-8001). For international inquiries (English-language), Dr. Masahide Hamaguchi will serve as liaison and interpreter (email: mhama@koto.kpu-m.ac.jp).

**Funding:** This work was supported by MHLW Comprehensive Research Project for Measures against Cardiovascular Diseases, Diabetes and Other Lifestyle Related Diseases Program Grant Number JPMH 24FA1008.

**Competing interests:** The authors have declared that no competing interests exist.

**Abbreviations:** BMI, body mass index; HbA1c, hemoglobin A1c; AST, aspartate aminotransferase; ALT, alanine aminotransferase; Cr, creatinine; eGFR, estimated glomerular filtration rate; T-C, total cholesterol; TG, triglycerides; HDL, high-density lipoprotein; MASLD, metabolic dysfunction-associated steatotic liver disease; MAFLD, metabolic dysfunction-associated fatty liver disease; NAFLD, non-alcoholic fatty liver disease; CKD, Chronic Kidney Disease.

an elevated risk of CKD compared to individuals without steatotic liver disease or cardiometabolic risk factors.

## Conclusions

This study establishes MASLD as a significant risk factor for CKD onset. Effective identification and management of MASLD cases are essential to mitigate the incidence of CKD.

## Introduction

The prevalence of obesity in Japan has surged due to the Westernization of dietary and lifestyle habits. This has led to an increase in non-alcoholic fatty liver disease (NAFLD), a condition primarily linked to visceral obesity and insulin resistance, often associated with metabolic syndrome. Globally, obesity-related diseases, including NAFLD, have become major health concerns due to their high prevalence and risks of progression to severe conditions such as liver cirrhosis, liver cancer, cardiovascular disease, and malignancies [1,2]. Numerous studies have investigated the relationship between NAFLD and the risk of developing type 2 diabetes [3]. The diagnosis of NAFLD hinges on the presence of hepatic steatosis, with other potential causes, such as viral hepatitis and substantial alcohol consumption, excluded. Predominant factors contributing to NAFLD include overnutrition, imbalanced diets, and physical inactivity, frequently accompanied by metabolic dysfunctions like obesity and glucose intolerance [4].

In May 2020, an International Consensus Panel redefined NAFLD as metabolic dysfunction-associated fatty liver disease (MAFLD), focusing on its metabolic underpinnings. MAFLD diagnosis requires evidence of fatty liver combined with one of the following: overweight/obesity, type 2 diabetes, or metabolic abnormalities in lean individuals. This reclassification highlights metabolic dysfunction as a central aspect of fatty liver disease, aligning it with contemporary clinical priorities [5,6], such as cardiometabolic risks. While long-term outcomes for MAFLD and NAFLD are similar, MAFLD includes metabolically healthy individuals, reflecting a heterogeneous population that requires careful evaluation. A cohort study in Japan involving 13,159 participants demonstrated MAFLD as an independent risk factor for chronic kidney disease (CKD) over a 10-year follow-up, with results adjusted for confounders such as eGFR, metabolic factors, and cardiovascular risks [7].

In 2023, a Delphi consensus proposed renaming steatotic liver diseases, replacing MAFLD with metabolic dysfunction-associated steatotic liver disease (MASLD). The term MASLD emphasizes inclusivity, addressing stigma associated with previous nomenclature, and accounts for diverse etiologies, such as "cryptogenic SLD (hepatic steatosis without cardiometabolic risk, irrespective of alcohol consumption,)" and "MetALD (MASLD with increased alcohol intake)". MASLD requires at least one cardiometabolic risk factor, including BMI, glucose, blood pressure, triglycerides, or HDL-C levels [8]. The multifactorial mechanisms connecting MASLD to CKD involve

interactions between renal and hepatic dysfunction [9]. This study aims to investigate MASLD's role as an independent risk factor for CKD and to identify the underlying associations and risk factors.

## Materials and methods

### Study design and participants

This longitudinal cohort study, known as the NAFLD in the Gifu Area, Longitudinal Analysis (NAGALA), was conducted to evaluate the effects of fatty liver on various components of metabolic syndrome starting in 1994. The current protocol received approval from the ethics committee (IRB number: 2022-04-01). Recruitment targeted individuals undergoing health checkups at Asahi University Hospital, inviting them to participate voluntarily. Waist circumference, which is essential to defining cardiometabolic risk, was only measured beginning in 2004, so the cohort comprised individuals attending these checkups from 1st January 2004–31st December 2023. The data were accessed for research purposes on 30th July 2024. Informed consent was obtained using an opt-out approach, as approved by the Ethics Committee. The requirement for written informed consent was waived due to the observational nature of the study and the use of de-identified data.

### Ethics statement

The study protocol was approved by the Ethics Committee of Asahi University Hospital (Approval No. 2022-04-01) and the study was conducted in accordance with the Declaration of Helsinki. In line with institutional policy for observational studies using de-identified data, the requirement for written informed consent was waived by the committee, and an opt-out approach by public notice was implemented for participants undergoing annual health checkups.

### Inclusion and exclusion criteria

Participants unwilling to participate, those with existing liver disease (e.g., viral hepatitis, autoimmune liver disease, and alcoholic hepatitis) or those with a history of medication use (e.g., corticosteroids and methotrexate) were excluded. Additional exclusions included individuals lacking baseline data on waist circumference, body mass index (BMI), abdominal ultrasound, high-density lipoprotein cholesterol (HDL-C), triglycerides (TG), blood pressure (systolic or diastolic), fasting plasma glucose, and those diagnosed with CKD at baseline.

### Data collection

Data were obtained through blood tests, abdominal ultrasonography, and questionnaires covering alcohol consumption, smoking habits, dietary patterns, and exercise routines. Weekly ethanol intake was calculated based on the reported quantity and type of alcoholic beverages. Regular exercisers were defined as individuals engaging in exercise at least once weekly.

### Definitions

**CKD.** CKD was defined as an eGFR < 60 mL/min/1.73㎡ or the presence of proteinuria (+1 to +3) persisting for a minimum of three months.

**Alcohol consumer.** Participants were categorized into three groups:

A-: Minimal alcohol consumers (<210 g/week for men, < 140 g/week for women)

A+: Light alcohol consumers (210–420 g/week for men, 140–350 g/week for women)

A++: Moderate – heavy alcohol consumers (>420 g/week for men, > 350 g/week for women)

**Grouping of SLD (A: alcohol, C: cardiometabolic risk, S: steatosis).** Based on the multisociety Delphi consensus [9], participants were classified into eight groups (see Figs 1,2):

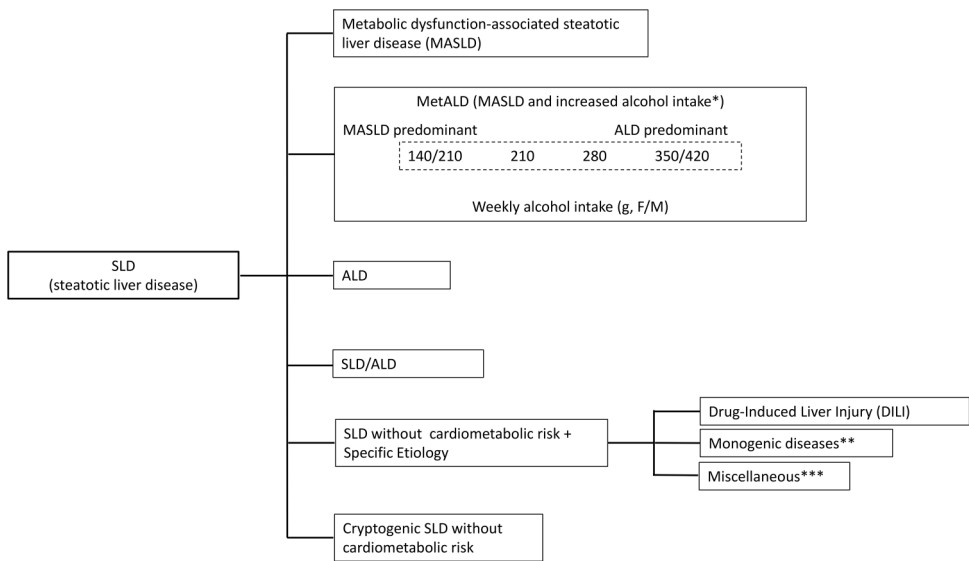

**Fig 1. Grouping of SLD.** This figure describes grouping of SLD. *Weekly intake 140-350g female, 210-420g male (average daily 20-50g female, 30-60g male). ** e.g. Lysosomal Acid Lipase Deficiency (LALD), hypobetalipoproteinemia, inborn errors of metabolism. *** e.g. Hepatitis C virus (HCV), malnutrition, celiac disease.

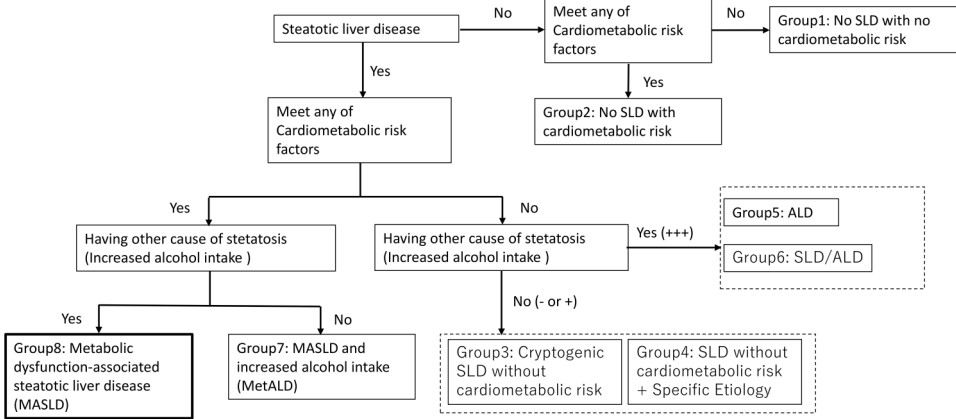

**Fig 2. Grouping of participants according to MASLD diagnosis.** Cardiometabolic risk factors: 1) BMI≧23 kg/m², 2) Fasting Plasma Glucose (FPG) ≧100 mg/dL or Hemoglobin A1c (HbA1c)≧5.7% or undergoing medication, 3) BP≧130/85 mmHg or use of antihypertensives, 4) TG≧150 mg/dL or use of antidyslipidemics, 5) HDL≦40 mg/dL in men and HDL≦50 mg/dL in women or use of antidyslipidemics. MASLD, metabolic dysfunction-associated steatotic liver disease; MetALD, MASLD and increased alcohol intake (Consumption of 140–350 g/week for females and 210–420 g/week for males); SLD, steatotic liver disease. ALD, alcohol-related liver disease (consumption of >350 g/week for females and >420 g/week for males); SLD without cardiometabolic risk + Specific etiology, those with hepatic steatosis, none of cardiometabolic risk factors. Specific etiology may means drug, virus, or monogenic disease. Cryptogenic SLD without cardiometabolic risk, those with hepatic steatosis, none of cardiometabolic risk factors.

- **Group 1 (A-/C-/S-):** No SLD, no CMR. Defined as individuals without hepatic steatosis (by abdominal ultrasonography), alcohol intake below thresholds, and no cardiometabolic risk factors (CMR): 1) BMI ≥ 23 kg/m², 2) fasting plasma glucose (FPG) ≥ 100 mg/dL or HbA1c ≥ 5.7%, 3) BP ≥ 130/85 mmHg, 4) TG ≥ 150 mg/dL, 5) HDL ≤ 40 mg/dL (men) or ≤ 50 mg/dL (women).

- **Group 2 (A-/C+/S-):** No SLD, with CMR. Same as Group 1, except with at least one CMR.

- **Group 3 (A- or +/C-/S+):** Cryptogenic SLD, without CMR. Includes hepatic steatosis without CMR, irrespective of alcohol consumption. In particular, steatosis derives from unknown origin.

- **Group 4 (A- or +/C-/S+):** SLD from specific etiology, without CMR. Includes hepatic steatosis from identifiable causes (e.g., drugs, viruses, monogenic disease) and no CMR.

- **Group 5 (A++/C-/S-):** Alcoholic liver disease (ALD). Defined as no hepatic steatosis, alcohol intake above thresholds, and no CMR.

- **Group 6 (A++/C-/S+):** SLD/ALD. Hepatic steatosis with alcohol intake above thresholds and no CMR.

- **Group 7 (A+/C+/S+):** MetALD (MASLD with alcohol intake). Hepatic steatosis with alcohol intake within light range and at least one CMR.

- **Group 8 (A-/C+/S+):** MASLD. Hepatic steatosis with alcohol intake below thresholds and at least one CMR.

### Main outcome

The primary outcome was CKD development during a five-year follow-up. Secondary analyses assessed odds risks of CKD across Groups 2–8 compared to Group 1.

### Statistical analysis

Analyses were conducted using JMP version 13.2 (SAS Institute, Cary, NC, USA). Results are reported as means with standard deviations (SD) or frequencies with percentages. Logistic regression models evaluated associations between SLD categories and CKD onset. The multivariate model adjusted for baseline age, sex, eGFR, smoking, and exercise. Results are expressed as odds ratios (ORs) with 95% confidence intervals (CIs), with p-values < 0.05 considered statistically significant.

## Results

As depicted in Fig 3, a total of 15,873 participants were included in this study. Table 1 outlines their baseline characteristics: 9,318 participants (58.7%) were male, with a mean age of 43.66 ± 8.43 years and a mean Body Mass Index (BMI) of 22.44 ± 3.38 kg/㎡. Among all participants, 6,094 individuals (38.4%) had a BMI exceeding 23 kg/㎡, and the mean waist circumference was 77.61 ± 9.50 cm. Smoking habits were reported by 3,674 participants (23.3%), while 2,584 individuals (16.4%) engaged in regular exercise. Regarding alcohol consumption, 14,119 participants (89%) were minimal alcohol consumers, 1,291 (8.1%) were light alcohol consumers, and 463 (2.9%) were moderate – heavy consumers.

Table 2 provides a detailed classification of participants into eight groups and their respective characteristics. Group 1 comprised 5,784 participants; Group 2, 6,581 participants; Group 3, 172 participants; Group 4, 2 participants; Group 5, 4 participants; Group 6, 125 participants; Group 7, 302 participants; and Group 8, 2,903 participants. Additionally, the table displays the incidence of CKD over a five-year follow-up. The highest CKD incidence was observed in Group 8 (9.5%), followed by Group 7 (8.3%), Groups 2 and 3 (both 7.8%), Group 6 (6.4%), and Group 1 (4.7%). No cases of CKD were reported in participants classified under Groups 4 or 5.

### Main outcome

In the multivariate analysis, the covariates included age, sex, baseline eGFR, exercise habits, and smoking status (Table 3). The adjusted odds ratios (ORs) for CKD incidence are detailed in Table 3, highlighting a significantly elevated CKD risk in Group

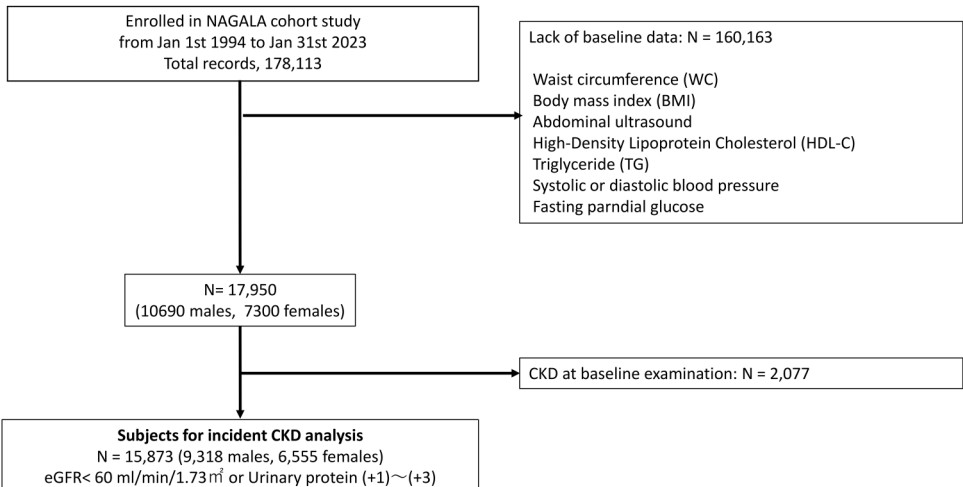

**Fig 3. Study flow of participants registration.** This figure shows the study flow of participant registration. A total of 15,873 participants were included in this study. NAGALA, NAfld in Gifu Area, Longitudinal Analysis; Incident CKD analysis.

8 compared to Group 1, which comprised individuals without steatosis or cardiometabolic risk factors. The OR for Group 2 (A-/C+/S-) was 1.20 (95% CI 0.98–1.37, p=0.07), while Group 3 (Cryptogenic SLD, A- or +/C-/S+) showed an OR of 1.50 (95% CI 0.76–2.47, p=0.30). For Group 6 (SLD/ALD, A+++/C-/S+), the OR was 0.77 (95% CI 0.39–1.63), and for Group 7, it was 1.10 (95% CI 0.70–1.74, p=0.67). Group 8 had a significantly increased OR of 1.37 (95% CI 1.12–1.67, p=0.002). Additionally, age (OR 1.04, 95% CI 1.03–1.05, p<0.001) and baseline eGFR (OR 0.88, 95% CI 0.87–0.89, p<0.001) emerged as significant predictors of CKD development. These findings confirm a statistically significant relationship between MASLD (Group 8) and an elevated CKD risk compared to Group 1.

Regarding alcohol consumption, we devided all participants into three groups. Minimal alcohol consumers (<210 g/week for males and 140 g/week for females), light alcohol consumers (210–420 g/week for males and 140–350 g/week for females), and moderate – heavy alcohol consumers (more than 420 g/week for males and more than 350 g/week for females).

## Discussion

Our study findings demonstrate that MASLD, as the newly defined category of steatotic liver disease (SLD), serves as an independent risk factor for the development of CKD over a five-year follow-up period. Previous research reported that more than 95% of individuals diagnosed with NAFLD met the criteria for MASLD [10]. Another study confirmed a substantially increased risk of CKD in a large, real-world cohort of adult NAFLD patients [11]. Based on this evidence, we hypothesize that MASLD is a significant contributor to CKD onset.

In contrast, MetALD did not exhibit a significant association with new CKD onset, although a trend toward statistical significance was observed (OR 1.10, 95% CI 0.70–1.74, p=0.67). Similarly, Group 2 (No SLD with cardiometabolic risk) showed a tendency toward CKD development (OR 1.16, 95% CI 0.98–1.37, p=0.07). These results suggest that cardiometabolic risk factors, rather than alcohol consumption, may play a more critical role in CKD development.

The relationship between alcohol consumption and kidney function has been explored in various studies with inconsistent conclusions. While alcohol intake exceeding 140 g/week has been associated with a greater than 30% decline in kidney function [12], daily consumption of 60 g or more has also been linked to increased kidney function decline [12]. Conversely, moderate alcohol intake (140–280 g/week) has been shown to slow CKD progression and

**Table 1. Charcteristics of all participants.**

| | n = 15,873 |
|---|---|
| Sex (male) | 9318 (58.7) |
| Age (yrs) | 43.66 ± 8.43 |
| Body weight (kg) | 62.05 ± 12.31 |
| Body mass index (kg/m²) | 22.44 ± 3.38 |
| Obesity (BMI ≧ 23 kg/m²) | 6094 (38.4) |
| Waist circumference (cm) | 77.61 ± 9.50 |
| HbA1c (%) | 5.29 ± 0.54 |
| Fasting plasma glucose(mg/dL) | 96.54 ± 15.23 |
| AST(IU/L) | 18.85 ± 12.32 |
| ALT(IU/L) | 21.17 ± 17.99 |
| Cr(mg/dL) | 0.78 ± 0.15 |
| eGFR(mL/min/1.73 m²) | 78.52 ± 12.37 |
| T-C(mg/dL) | 198.13 ± 33.08 |
| TG(mg/dL) | 83.53 ± 69.38 |
| HDL-C(mg/dL) | 57.78 ± 16.44 |
| Systolic blood pressure(mmHg) | 116.45 ± 15.40 |
| Diastolic blood pressure(mmHg) | 72.58 ± 11.01 |
| Cigarette(Current-smoker) | 3674 (23.3) |
| Cigarette(Ex-smoker) | 3332 (21.1) |
| Minimal alcohol consumer(M: < 210 g/w, F: < 140 g/w) | 14119 (89.0) |
| Light alcohol consumer (M: 210–420 g/w, F: 140–350 g/w) | 1291 (8.1) |
| Moderate – heavy alcohol consumer (M: 420 > g/w, F: > 350 g/w) | 463 (2.9) |
| Exercise (≧1day/w) | 2584 (16.4) |

This table shows the characteristics of all participants. The characteristics of all participants. 9,318 were men (58.7%), and the mean age was 43.66 ± 8.43 years, and the mean Body Mass Index (BMI) was 22.44 ± 3.38 kg/㎡. 6,094 people (38.4%) had BMI of over 23 kg/㎡. Mean waist circumstance was 77.61 ± 9.50 cm. 3,674 (23.3%) had smoking habits and 2,584 (16.4%) had exercise routine. 14,119 (89%) were minimal alcohol consumers, 1,291 (8.1%) were light consumers, and 463 (2.9%) were moderate – heavy consumers. Obesity is defined as BMI > 23 kg/m². BMI, body mass index; HbA1c, hemoglobin A1c; AST aspartate aminotransferase; ALT alanine aminotransferase; Cr Creatinine; eGFR, estimated glomerular filtration rate; T-C Total Cholesterol; TG Triglycerides; HDL, high-density lipoprotein.

even reduce the incidence of cardiovascular disease (CVD) [13]. Light alcohol consumption (<140 g/week) has been associated with lower levels of inflammatory markers and a reduced risk of CKD [14]. The effects of light alcohol consumption (up to 280 g/week) on kidney function remain unclear, and the risk of CKD in individuals with higher alcohol consumption (Group 5: ALD, Group 6: ALD/SLD) could not be conclusively determined due to the small number of participants and the absence of cardiometabolic risk factors. Further research is needed to clarify these relationships.

As anticipated, age and baseline eGFR were significantly associated with CKD incidence. NAFLD is predominantly observed in obese individuals in Western populations, while non-obese NAFLD is more prevalent in Asia, including Japan [15]. In non-obese individuals with NAFLD, factors such as increased visceral fat, reduced muscle mass and strength, muscle atrophy (pre-sarcopenia), and impaired glucose tolerance are key pathophysiological contributors [15]. In Japan, individuals with lean NAFLD are typically older and have the highest all-cause mortality, followed by those with overweight

**Table 2. Characteristics and CKD incidence of 8 groups.**

| | Group1 no SLD no Cardiometabolic risk | Group2 no SLD with Cardiometabolic risk | Group3 Cryptogenic SLD without Cardiometabolic risk | Group4 SLD without Cardiometabolic risk + Specific Etiology | Group5 ALD | Group6 SLD/ALD | Group7 MetALD | Group8 MASLD |
|---|---|---|---|---|---|---|---|---|
| n | 5,784 | 6,581 | 172 | 2 | 4 | 125 | 302 | 2,903 |
| A/C/S | A-/C-/S- | A-/C+/S- | A- or +/C-/S+ | A- or+/C-/S+ | A+++/C-/S- | A+++/C-/S+ | A+/C+/S+ | A-/C+/S+ |
| Sex (male) | 2225 (38.5) | 4158 (63.2) | 139 (80.8) | 2 (100) | 2 (50) | 123 (98.4) | 290 (96.0) | 2379 (82.0) |
| Age (yrs) | 41.1±7.4 | 45.1±8.8 | 43.1±8.4 | 44.5±10.6 | 49.3±6.7 | 46.7±8.6 | 46.6±7.9 | 45.1±8.4 |
| Body weight (kg) | 53.7±7.7 | 63.1±10.2 | 61.1±6.8 | 60.9±1.7 | 60.9±10.8 | 75.6±11.3 | 75.3±9.9 | 74.3±11.8 |
| Body mass index (kg/ m$^2$) | 20.0±1.7 | 22.8±2.7 | 21.6±1.1 | 21.4±0.2 | 21.3±1.1 | 25.8±3.1 | 25.8±2.8 | 26.2±3.4 |
| Obesity (BMI≧23 kg/ m$^2$) | 0 | 3356 (51.0) | 0 | 0 | 0 | 102 (81.6) | 269 (89.1) | 2498 (86.0) |
| Waist circumference (cm) | 70.6±5.8 | 78.7±7.6 | 77.3±4.3 | 78.8±2.5 | 77.9±8.9 | 88.1±7.3 | 88.5±7.6 | 87.7±8.2 |
| HbA1c (%) | 5.1±0.3 | 5.3±0.5 | 5.2±0.3 | 5.2±0.6 | 4.9±0.2 | 5.5±0.7 | 5.5±0.8 | 5.6±0.8 |
| Fasting plasma glucose(mg/dL) | 89.4±5.7 | 98.0±13.5 | 93.0±4.8 | 94.5±6.4 | 88.8±1.5 | 111.7±23.8 | 108.1±23.7 | 105.8±22.1 |
| AST(IU/L) | 16.7±7.1 | 18.4±15.6 | 19.4±7.0 | 17.0±7.1 | 23.0±2.0 | 27.1±13.3 | 25.0±11.9 | 23.1±10.8 |
| ALT(IU/L) | 15.4±7.8 | 19.6±19.6 | 26.4±14.6 | 17.5±2.1 | 23.3±3.9 | 32.4±15.1 | 33.4±20.8 | 34.3±21.0 |
| Cr(mg/dL) | 0.73±0.14 | 0.80±0.15 | 0.85±0.15 | 0.81±0.0 | 0.73±0.15 | 0.88±0.10 | 0.87±0.11 | 0.85±0.14 |
| eGFR(mL/min/ 1.73 m$^2$) | 81.1±13.1 | 77.5±11.9 | 76.8±11.7 | 83.2±6.5 | 79.3±7.9 | 74.9±10.1 | 75.8±10.8 | 76.3±11.3 |
| T-C(mg/dL) | 190.8±30.0 | 198.7±33.9 | 202.6±32.1 | 223±11.3 | 204.3±40.9 | 213.0±31.3 | 209.5±33.8 | 209.6±33.2 |
| TG(mg/dL) | 52.8±25.2 | 85.7±62.3 | 83.3±31.0 | 68.5±20.5 | 88.3±32.1 | 169.1±106.3 | 156.4±179.3 | 128.7±87.1 |
| HDL-C(mg/dL) | 66.2±14.6 | 55.8±16.4 | 55.1±12.4 | 52.5±7.8 | 60.2±13.2 | 53.5±15.3 | 49.5±13.0 | 46.7±11.3 |
| Systolic blood pressure(mmHg) | 107.2±10.5 | 119.7±15.1 | 112.2±8.1 | 109.3±6.0 | 104.4±8.0 | 129.9±14.4 | 129.4±13.5 | 126.0±14.6 |
| Diastolic blood pressure(mmHg) | 66.1±7.9 | 74.8±10.8 | 70.0±6.0 | 68.8±1.1 | 64.9±6.4 | 82.4±9.3 | 83.0±9.4 | 79.2±10.2 |
| Cigarette(Current-smoker) | 978 (17.0) | 1737 (26.5) | 31 (18.1) | 1 (50.0) | 1 (25.0) | 48 (38.7) | 121 (40.7) | 757 (21.0) |
| Cigarette (Ex-smoker) | 826 (14.3) | 1472 (22.5) | 43 (25.2) | 1 (50.0) | 3 (75.0) | 61 (49.2) | 124 (41.8) | 802 (24.1) |
| Minimal alcohol consumer(M:<210 g/w, F:<140 g/w) | 5376 (92.9) | 5673 (86.2) | 165 (95.9) | 2 (100) | 0 | 0 | 0 | 2903 (100) |
| Light alcohol consumer (M: 210–420 g/w, F: 140–350 g/w) | 310 (5.4) | 674 (10.2) | 7 (4.1) | 0 | 0 | 0 | 302 (100) | 0 |
| Moderate – heavy alcohol consumer (M: 420>g/w, F:>350 g/w) | 98 (1.7) | 234 (3.6) | 0 | 0 | 4 (100) | 125 (100) | 0 | 0 |
| Exercise (≧1day/w) | 943 (16.4) | 1148 (17.6) | 28 (16.3) | 0 (0) | 1 (25.0) | 24 (19.2) | 47 (15.6) | 393 (13.7) |
| Incident CKD (%) | 273 (4.7) | 513 (7.8) | 14 (7.8) | 0 | 0 | 8 (6.4) | 25 (8.3) | 275 (9.5) |

Table 2 describes the classification of 8 groups and characteristics of each group.

Group 1 comprised 5,784 participants; Group 2, 6,581 participants; Group 3, 172 participants;

Group 4, 2 participants; Group 5, 4 participants; Group 6, 125 participants; Group 7, 302 participants;

and Group 8, 2,903 participants. The highest incidence of CKD was observed in Group 8 (9.5%).

**Table 3. Odds ratios of incident CKD according to the 8 groups of fatty liver disease.**

|  | p-value | OR(95%CI) |
|---|---|---|
| Age | <0.001 | 1.04 (1.03–1.05) |
| Sex | 0.02 | 1.03 (0.88–1.21) |
| eGFR at baseline | <0.001 | 0.88 (0.87–0.89) |
| Exercise | 0.07 | 1.16 (0.99–1.36) |
| Cigarette (Ex-smoker) | 0.05 | 1.20 (1.00–1.44) |
| Cigarette (Current-smoker) | 0.627 | 1.15 (0.95–1.38) |
| Group2(A-/C+/S-) | 0.07 | 1.16 (0.98–1.37) |
| Group3(Cryptogenic SLD, A- or +/C-/S+) | 0.3 | 1.50 (0.76–2.47) |
| Group4(Specific etiology, A- or +/C-/S+) | 0.99 | 6.28E-07 |
| Group5(ALD, A+++/C-/S-) | 0.99 | 2.20E-07 |
| Group6(SLD/ALD, A+++/C-/S+) | 0.49 | 0.77 (0.39–1.63) |
| Group7(MetALD, A+/C+/S+) | 0.67 | 1.10 (0.70–1.74) |
| Group8(MASLD, A-/C+/S+) | 0.002 | 1.37 (1.12–1.67) |

This table shows the result of multivariate analysis. Covariates were age, sex, eGFR at baseline, exercise, and smoking status. Group 8 presented an OR of 1.37 (95% CI 1.12–1.67, p=0.002). Age (OR 1.04, 95% CI 1.03–1.05, p<0.001) and eGFR (OR 0.88, 95% CI 0.87–0.89, p<0.001) were also significant risk factors for the development of CKD.

and obese NAFLD [16]. Given the similarities in MASLD, eGFR might be overestimated in this population, potentially underestimating CKD risk. Measuring cystatin C, which is unaffected by muscle mass, could improve CKD risk assessment in MASLD patients. Personalized approaches, including medical nutritional therapy and exercise, may be crucial for individuals at higher risk.

This study has several limitations. The participants were individuals undergoing annual health checkups, limited to the Gifu region in Japan, introducing potential selection bias. Additionally, participants who developed new internal medical conditions during the study might have dropped out. Dietary data beyond alcohol intake, such as the consumption of low-fat, low-sugar dairy products (which may protect against MASLD) or the role of plant-based protein diets in CKD, were not collected [17,18]. Although we recorded fasting plasma glucose levels, postprandial glucose measurements were unavailable. Studies have shown that impaired glucose tolerance (IGT), but not impaired fasting glucose (IFG), is a risk factor for cardiovascular disease [19–21]. These findings, collected since 1994, underline the need for future studies to address these gaps.

## Conclusion

MASLD was identified as a significant independent risk factor for CKD onset. Early identification and management of MASLD cases are critical to mitigating the risk of developing CKD.

## Acknowledgments

We thank all staff members of the Kyoto Prefectural University of Medicine and Asahi University Hospital.

## Author contributions

**Data curation:** Kimiko Sakai.

**Supervision:** Masahide Hamaguchi, Takuro Okamura, Hiroshi Okada, Akihiro Obora, Takao Kojima, Michiaki Fukui.

**Writing – original draft:** Yuriko Ono.

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
