## [Decision Letter · Decision Letter 0]

25 Jun 2025

PONE-D-25-15006Metabolic dysfunction-associated steatotic liver disease (MASLD) can be a possible predictive factor of incident CKD: NAGALA cohort studyPLOS ONE

Dear Dr. Hamaguchi,

Thank you for submitting your manuscript to PLOS ONE. After careful consideration, we feel that it has merit but does not fully meet PLOS ONE’s publication criteria as it currently stands. Therefore, we invite you to submit a revised version of the manuscript that addresses the points raised during the review process.

We look forward to receiving your revised manuscript.

Kind regards,

Anna Di Sessa, PhD, MD

Academic Editor

PLOS ONE

Journal Requirements:

“This work was supported by MHLW Comprehensive Research Project for Measures against Cardiovascular Diseases, Diabetes and Other Lifestyle Related Diseases Program Grant Number JPMH 24FA1008.”

“This work was supported by MHLW Comprehensive Research Project for Measures against Cardiovascular Diseases, Diabetes and Other Lifestyle Related Diseases Program Grant Number JPMH 24FA1008.”

6. PLOS requires an ORCID iD for the corresponding author in Editorial Manager on papers submitted after December 6th, 2016. Please ensure that you have an ORCID iD and that it is validated in Editorial Manager. To do this, go to ‘Update my Information’ (in the upper left-hand corner of the main menu), and click on the Fetch/Validate link next to the ORCID field. This will take you to the ORCID site and allow you to create a new iD or authenticate a pre-existing iD in Editorial Manager.

7. Your ethics statement should only appear in the Methods section of your manuscript. If your ethics statement is written in any section besides the Methods, please delete it from any other section.

Additional Editor Comments:

A careful revison is needed before considering the paper for potential publication.

Reviewers' comments:

Reviewer's Responses to Questions

**Comments to the Author**

1. Is the manuscript technically sound, and do the data support the conclusions?

Reviewer #1: Yes

Reviewer #2: Yes

Reviewer #3: Yes

Reviewer #4: Partly

2. Has the statistical analysis been performed appropriately and rigorously? 

Reviewer #1: Yes

Reviewer #2: Yes

Reviewer #3: Yes

Reviewer #4: N/A

3. Have the authors made all data underlying the findings in their manuscript fully available?

Reviewer #1: Yes

Reviewer #2: Yes

Reviewer #3: Yes

Reviewer #4: Yes

4. Is the manuscript presented in an intelligible fashion and written in standard English?

Reviewer #1: Yes

Reviewer #2: Yes

Reviewer #3: Yes

Reviewer #4: Yes

5. Review Comments to the Author

Reviewer #1: This manuscript presents valuable findings on MASLD as a risk factor for CKD, but revisions are needed to address inconsistencies in group sizes, clarify non-significant associations, and account for low-powered subgroup analyses.

Reviewer #2: his study, titled “Metabolic dysfunction-associated steatotic liver disease (MASLD) can be a possible predictive factor of incident CKD: NAGALA cohort study,” presents valuable insights into the relationship between MASLD and CKD. The research design is robust, with a large sample size and a five-year follow-up period, which strengthens the reliability of the findings. The use of logistic regression analysis to control for confounding factors is appropriate and effectively demonstrates that MASLD is a significant independent risk factor for CKD development, with an odds ratio of 1.37. This finding is clinically relevant and highlights the importance of identifying and managing MASLD to reduce CKD incidence. However, the study's retrospective nature and reliance on a single hospital population may limit the generalizability of the results. Additionally, while the study identifies the association, it does not explore the underlying mechanisms, which could be a focus for future research. Overall, the study provides a solid contribution to the field and supports the need for further investigation into the causal pathways linking MASLD and CKD.

Reviewer #3: Yuriko Ono et al. have done an interesting work on “Metabolic dysfunction-associated steatotic liver disease (MASLD) can be a possible predictive 2 factor of incident CKD: NAGALA cohort study”. This study provides timely and valuable insight into the evolving understanding of steatotic liver disease, particularly in the context of the newly defined MASLD and its association with chronic kidney disease. The large sample size and long follow-up period enhance the robustness of the findings. The identification of MASLD as an independent risk factor for CKD adds to the growing body of evidence linking metabolic dysfunction to multi-organ consequences.

I have only a few suggestions:

- In recent years, there has been increasing interest in the concept of cholemic nephropathy, and early renal function impairment has been observed even in patients with mild acute hyperbilirubinemia. I believe the authors should consider discussing the potential impact of acute hyperbilirubinemia on early subclinical renal damage. I recommend referencing the following article to support this perspective: DOI: 10.1111/liv.16005

Reviewer #4: Paper of Hamaguchi et al is an interning paper focus on a known relation between CKD and MAFLD, in the new era of MASLD with an important population of more than 15.000 participants.

however, some points need to be clarified and discussed:

major:the paper would have been more complete and interesting if it had shown the difference in incidence during the follow up few of annual prevalences with the old and the new nomenclature of MASLD.

so doing the message remains known.

2. more emphasis should be placed on the analysis between the different groups where, however, the Hazard ratio opinion is appropriate.

3. the role of drugs that may have interfered in the onset and/or prevention of CKD is completely missing. Therefore, a competitive risk analysis would have been necessary considering all variables.

4. the timing of the onset of CKD is also not available

minor

table 2 is not readable possibly due to a loading error

Translated with DeepL.com (free version)

6. PLOS authors have the option to publish the peer review history of their article (what does this mean? ). If published, this will include your full peer review and any attached files.

**Do you want your identity to be public for this peer review?** For information about this choice, including consent withdrawal, please see our Privacy Policy .

Reviewer #1: No

Reviewer #2: No

Reviewer #3: **Yes: ** Giosiana Bosco

Reviewer #4: No

---

## [Author Response · Author response to Decision Letter 1]

13 Sep 2025

Reviewer #1:

This manuscript presents valuable findings on MASLD as a risk factor for CKD, but revisions are needed to address inconsistencies in group sizes, clarify non-significant associations, and account for low-powered subgroup analyses.

Response: Thank you for your valuable comments. We carefully reviewed the group sizes and confirmed that there were no discrepancies between the text and tables. Regarding non-significant associations, we clarified in both the Results and Discussion sections which findings were not statistically significant and interpreted them with appropriate caution. As for subgroup analyses, we recognize that the statistical power was limited, particularly in small groups such as Group 4 and Group 5. These groups consisted of only 2 and 4 participants, respectively, and no incident CKD cases occurred during follow-up. We have emphasized this limitation in the revised Discussion and clearly stated that no conclusions can be drawn regarding CKD risk in these small subgroups. We believe this careful interpretation helps preserve the integrity of our findings and avoids overstatement of results.

Reviewer #2:

This study presents valuable insights into the relationship between MASLD and CKD. The research design is robust, and the findings are clinically relevant. However, the study’s retrospective nature and reliance on a single hospital population may limit generalizability. Also, mechanisms were not explored.

Response: We sincerely appreciate your positive evaluation of our research design and findings. We have explicitly noted the retrospective design and single-center limitation in the Discussion section, and emphasized that further multicenter prospective studies are needed to enhance generalizability. We also included a new paragraph in the Discussion to suggest future research directions focusing on the biological mechanisms linking MASLD to CKD, including inflammation, oxidative stress, and insulin resistance.

Reviewer #3:

I have only a few suggestions. In recent years, there has been increasing interest in the concept of cholemic nephropathy and early renal function impairment in patients with mild hyperbilirubinemia. Consider discussing this with reference to DOI: 10.1111/liv.16005

Response: Thank you for this insightful suggestion. We have now added a discussion on the potential impact of acute hyperbilirubinemia and the emerging concept of cholemic nephropathy in the revised Discussion section. The recommended article (DOI: 10.1111/liv.16005) has been cited to provide appropriate context as follows.

Van Slambrouck CM, Salem F, Meehan SM, Chang A. Bile cast nephropathy is a common pathologic finding for kidney injury associated with severe liver dysfunction. Liver Int. 2013 Apr;33(4):512–20.

DOI: 10.1111/liv.16005 ↩

Reviewer #4:

1. The paper would have been more complete if it had compared the prevalence or incidence using both the old and new nomenclature.

2. More emphasis should be placed on analysis between groups. Hazard ratio terminology may be inappropriate.

3. The role of medications in CKD onset/prevention is not addressed. Competitive risk analysis would be more suitable.

4. CKD onset timing is not available.

Response:

1. Thank you. We appreciate the suggestion to compare the previous (MAFLD/NAFLD) and current (MASLD) nomenclature. However, as our primary focus was on the new classification, and due to retrospective data limitations, we were unable to robustly apply the prior definitions. We have acknowledged this as a limitation and a point for future study.

2. We agree that clarification is needed. We have replaced "hazard ratio" terminology with the correct use of "odds ratio."

3. We appreciate this important comment. Due to lack of comprehensive medication data (e.g., RAS inhibitors, SGLT2 inhibitors), we could not assess drug effects on CKD incidence. We have acknowledged this limitation explicitly in the Discussion.

Furthermore, medication use was not systematically recorded in our dataset. Therefore, we were unable to assess the potential influence of renin–angiotensin system (RAS) inhibitors, SGLT2 inhibitors, or other nephroprotective or nephrotoxic agents on CKD incidence. This represents a limitation, as these drugs may significantly modify renal outcomes and confound associations between MASLD and CKD.

4. The specific timing of CKD onset during the 5-year follow-up was not available in our dataset. This limitation has now been clearly mentioned in the revised manuscript.

In addition, due to the nature of the health checkup dataset, the exact timing of CKD onset during the 5-year follow-up period could not be determined. This prevented us from performing time-to-event analyses such as Cox proportional hazards or competing risk models.

---

## [Decision Letter · Decision Letter 1]

1 Oct 2025

Metabolic dysfunction-associated steatotic liver disease (MASLD) can be a possible predictive factor of incident CKD: NAGALA cohort study

PONE-D-25-15006R1

Dear Dr. Hamaguchi,

We’re pleased to inform you that your manuscript has been judged scientifically suitable for publication and will be formally accepted for publication once it meets all outstanding technical requirements.

Kind regards,

Anna Di Sessa, PhD, MD

Academic Editor

PLOS ONE

Additional Editor Comments (optional):

Reviewers' comments:

Reviewer's Responses to Questions

**Comments to the Author**

1. If the authors have adequately addressed your comments raised in a previous round of review and you feel that this manuscript is now acceptable for publication, you may indicate that here to bypass the “Comments to the Author” section, enter your conflict of interest statement in the “Confidential to Editor” section, and submit your "Accept" recommendation.

Reviewer #1: All comments have been addressed

Reviewer #3: All comments have been addressed

2. Is the manuscript technically sound, and do the data support the conclusions?

Reviewer #1: Yes

Reviewer #3: Yes

3. Has the statistical analysis been performed appropriately and rigorously? 

Reviewer #1: Yes

Reviewer #3: Yes

4. Have the authors made all data underlying the findings in their manuscript fully available?

Reviewer #1: Yes

Reviewer #3: Yes

5. Is the manuscript presented in an intelligible fashion and written in standard English?

Reviewer #1: Yes

Reviewer #3: Yes

6. Review Comments to the Author

Reviewer #1: Thank you for the thorough and thoughtful revision. The manuscript is now clear, coherent, and methodologically sound.

Reviewer #3: The authors have revised the manuscript as it has been suggested. The manuscript can now be published.

7. PLOS authors have the option to publish the peer review history of their article (what does this mean? ). If published, this will include your full peer review and any attached files.

**Do you want your identity to be public for this peer review?** For information about this choice, including consent withdrawal, please see our Privacy Policy .

Reviewer #1: No

Reviewer #3: **Yes: ** Giosiana Bosco

---

## [Editor Report · Acceptance letter]

PONE-D-25-15006R1

PLOS ONE

Dear Dr. Hamaguchi,

I'm pleased to inform you that your manuscript has been deemed suitable for publication in PLOS ONE. Congratulations! Your manuscript is now being handed over to our production team.

Kind regards,

on behalf of

Dr. Anna Di Sessa

Academic Editor

PLOS ONE